# Makerspaces in First-Year Engineering Education

**Pooya Taheri [1,2,*] , Philip Robbins [3] and Sirine Maalej [1]**

[1]   Computing Science & Information Systems Department, Langara College, Vancouver, BC V5Y 2Z6, Canada; smaalej@langara.ca

[2]   Mechatronic Systems Engineering Department, Simon Fraser University, Surrey, BC V3T 0A3, Canada

[3]   Fine Arts Department, Langara College, Vancouver, BC V5Y 2Z6, Canada; probbins@langara.ca

*   Correspondence: ptaherig@sfu.ca

**Abstract:** Langara College, as one of the leading undergraduate institutions in the province of British Columbia (BC), offers the "Applied Science for Engineering" two-year diploma program as well as the "Engineering Transfer" two-semester certificate program. Three project-based courses are offered as part of the two-year diploma program in Applied Science (APSC) and Computer Science (CPSC) departments: "APSC 1010—Engineering and Technology in Society", "CPSC 1090—Engineering Graphics", and "CPSC 1490—Applications of Microcontrollers", with CPSC 1090 and CPSC 1490 also part of the Engineering Transfer curriculum. Although the goals, scopes, objectives, and evaluation criteria of these courses are different, the main component of all three courses is a group-based technical project. Engineering students have access to Langara College's Makerspace for the hands-on component of their project. Makerspaces expand experiential learning opportunities and allows students to gain a skillset outside the traditional classroom. This paper begins with a detailed review of the maker movement and the impact of makerspace in higher education. Different forms of makerspace and the benefits of incorporating them on first-year students' creativity, sense of community, self-confidence, and entrepreneurial skills are discussed. This paper introduces Langara's engineering program and its project-based design courses. Langara's interdisciplinary makerspace, its goals and objectives, equipment, and some sample projects are introduced in this paper in detail. We then explain how the group-project component of APSC 1010, CPSC 1090, and CPSC 1490 are managed and how using makerspace improves students' performance in such projects. In conclusion, the paper describes the evaluation of learning outcomes via an anonymous student survey.

**Keywords:** makerspace; engineering education; project-based learning; first-year higher education

## 1. Introduction

### 1.1. What Is a Makerspace?

The makerspace model, originating in the DIY (do-it-yourself) community, is among the most prominent keywords in Science, Technology, Engineering, and Math (STEM) education today [1]. Makerspaces are, in general, physical spaces with shared resources used to pursue technical projects with the support of a maker community. Drivers of the maker movement include inexpensive technology, open-source hardware and software, as well as globalization. Makerspace centres commonly include woodworking, electronics, robotics, and fabrication, with making activities including [2]:

1.   Circuits, switches, and electricity;
2.   Textiles and soft circuits;
3.   Robots, motors, and mechanics;
4.   Rockets and flight;

5. Deconstruction and construction;
6. Carpentry and architecture;
7. 3D printing;
8. Programming.

The central pillar of makerspaces is hands-on prototyping and designing. Both of these skills are shown to play a key role in the development of students' creativity and the visceral understanding of concepts [3]. Makerspace—as a space to make choices, to create, to connect, and to support technology, innovation, and entrepreneurship—aims to promote a sense of community-based participatory culture that encourages informal interactions and peer education [4–6].

The combination of equipment and culture provides a powerful catalyst for formal and informal self-sustaining learning culture that fosters imagination and encourages cooperation [7]. Makerspaces (also referred to in the literature as Hackerspaces, HackLabs, FabLabs, and TechShops) can be classified into different categories based on their foci [8]:

1. Industrial/Technical Makerspace;
2. Commercial Makerspace;
3. Educational Makerspace;
4. Community/Library Makerspace;
5. Hub/Network Makerspace.

*1.2. Makerspace History*

The last 150 years have seen an increasingly heavy emphasis on theory and mathematical modeling as opposed to a more practice-based approach in the engineering curricula. This emphasis is partially due to the increasing depth of engineering science and partially due to the difficulty of managing shop resources for a growing number of students. As skills beyond theoretical knowledge—such as interpersonal skills, problem-solving, and critical thinking—grew in demand by industry, frustration also grew regarding the graduating students' lack of hands-on skills [3,9].

The maker movement originated in the form of the Fabrication Labs (FabLabs) used for engineering education at MIT in 2001. FabLab was designed to support the course "How to Make (Almost) Anything", and intended to familiarize students with digital fabrication and rapid prototyping [10,11]. The FabLab concept was meant to provide basic, accessible, and low-cost fabrication capabilities. Common FabLab equipment included 3D printers, CNC (Computer Numerical Control) mills/lathes, Printed Circuit Board (PCB) milling/etching equipment, CNC cutting systems, and microprocessor/digital electronics equipment. MIT's FabLab also provided peer-to-peer training to leverage the fabrication skills of members [7].

Another important development that fuelled the makerspace movement was the increased ease of access to information, including equipment/tool training, tutorials, and supplies (such as motors, materials, sensors, electronic components, and fasteners, to name but a few). With the publication of MAKE magazine in 2011, makerspaces became even more popular in schools, colleges, universities, libraries, and industries.

Makerspaces are now considered one of the most important developments in engineering and design education. Students use their creativity and critical-thinking skills to engage in more hands-on projects with the purpose of finding suitable solutions to real-world design problems [12]. The process of making in makerspaces fosters the ability to apply technical knowledge, to work in intercultural teams, and to build knowledge and skills independently. It also serves to grow a culture of community-based collaboration of helping others and sharing expertise [4,12].

*1.3. Makerspace Goals*

Makerspaces as holistic places of learning offer an excellent space to conceive, design, implement, and build prototypes that are project-based, interdisciplinary, and solve real world problems [13].

By fostering creativity and an awareness of the importance of trial and error, programs that incorporate makerspace education have shown a significant increase in student participation [9,14]. Courses taught in higher-education makerspaces meet long-standing design education goals [15,16].

The following educational impacts have been observed in higher-education makerspaces [9,17]:

1. Improving students' communication, collaboration, and teamwork skills;
2. Developing problem-solving and investigation skills;
3. Introducing students to the design cycle;
4. Nurturing entrepreneurial, leadership, and management skills;
5. Increasing students' self-confidence;
6. Providing opportunities for hands-on and technical experience [4];
7. Demonstrating manufacturing/prototyping methods and the challenges/limitations involved;
8. Preparing students for professional careers [10];
9. Improving students' academic success and GPAs [18,19];
10. Establishing connections and partnerships between engineering disciplines and society;
11. Providing opportunities for self-directed learning, volunteering, and peer education [20].

Based on previous research, to maximize the makerspace's impact on the quality of education, many key points must be considered [21]:

1. The mission of the academic makerspace must be clearly defined from the onset;
2. The facility must be properly staffed with educators, manufacturing and design professionals, as well as administrative support [22];
3. Access times must be aligned with the students' work schedules;
4. Providing user trainings and workshops is essential;
5. Attention must be devoted to establishing a co-maker community on campus;
6. Outfitting, training, safety, financing, and staffing models should be fully developed;
7. Special attention needs to be taken towards design aspects to draw the student population into the grassroots, community-formed makerspaces [11];
8. Tidiness, familiarity, and the optimal use of the design makerspaces offer a more-welcoming environment [23];
9. The ongoing maintenance and tooling cost of fabrication equipment should be scheduled in a consistent manner [9];
10. A pedagogical design should be designed to stimulate students' innovative skills while respecting the formal requirements [24].

### 1.4. Objective and Structure

In most engineering programs, technical and science courses are offered along with labs and workshop sessions. The scope of these sessions is particularly focused on the specific technical course details. In this paper, based on the quality of course projects, instructors' observations, and student evaluations, it is proposed that early introduction of makerspace can be a promising factor to improve students' active learning throughout an engineering program.

Langara College's "Applied Science for Engineering" two-year diploma and the "Engineering Transfer" two-semester certificate programs offer a structured curriculum for the students interested in engineering disciplines to gain the theoretical and practical knowledge equivalent to the first-year of an engineering Bachelor's program. These programs help students transfer to their field of interest with a better understanding of engineering disciplines. Unlike four-year programs where students are introduced to group projects in their final years of studies, Langara's engineering diploma program offers three project-based courses to first- and second-year students listed below:

1. APSC 1010—Engineering and Technology in Society;
2. CPSC 1090—Engineering Graphics;
3. CPSC 1490—Applications of Microcontrollers.

Since 2017, Langara's Makerspace has been utilized to facilitate the project component of the project-based engineering courses in addition to multiple courses from other disciplines. This paper is based on the authors' experiences on the impact of using makerspace in teaching APSC 1010, CPSC 1090, and CPSC 1490. Throughout this paper we focus on the benefits of general makerspaces for the first-year students who are interested in engineering but are not necessarily familiar with the specifics of engineering disciplines. This paper demonstrates how utilizing makerspace can solidify the students' sense of community while improving their hands-on technical skills. It is also shown that makerspace assists engineering education in nurturing confidence, increasing retention, and improving recruitment of students in their fields of interest.

The paper begins with a detailed review of the maker movement and the impact of makerspace in higher education. It also introduces the project-based design courses in Langara's engineering program. In conclusion, the paper describes the evaluation of learning outcomes via an anonymous student survey.

The abbreviations used in this paper are listed in Table 1.

**Table 1.** Nomenclature.

| | |
|---|---|
| APSC | Applied Science |
| BC | British Columbia |
| CAD | Computer-Aided Design |
| CNC | Computer Numerical Control |
| CPBL | Cooperative Project-Based Learning |
| CPSC | Computer Science |
| DIY | Do It Yourself |
| EML | Entrepreneurial-Minded Learning |
| FabLab | Fabrication Lab |
| GPA | Grade Point Average |
| IoT | Internet of Things |
| SET | Student Evaluation of Teaching |
| STEM | Science, Technology, Engineering, and Math |

## 2. Makerspace Considerations and Benefits

### 2.1. Makerspace Safety

As the safety of students and staff is of paramount importance, the development of stringent safety protocols is necessary to ensure the proper functioning of the makerspace [9]. Workshops and protocols are organized to keep the equipment functional and mitigate the loss of makerspace knowledge after student graduation. This safety culture encourages and enforces personal accountability and promotes dialogue among community members [7,10,25]. Makerspace encourages safe experimentation to build students' confidence while practicing peer-to-peer community-based training [1].

### 2.2. Diversity in Makerspaces

While the number of female students enrolled in engineering and STEM programs are increasing, women, particularly women of color, are still more likely than their male peers to leave engineering majors. Several factors contribute to this result, including reduced self-efficacy, the stereotype threat, deeply ingrained masculine culture, disappointment with courses, and a lack of community in engineering settings [26]. Makerspace education has the potential to empower female K–12 and undergraduate students by providing a place where students can identify as engineers. Makerspace

also provides opportunity for community-based engagement by supporting relationships between students, their engineering peers, and leaders [26–29].

People with disabilities and youth growing up in poverty are also largely underrepresented in STEM fields [30]. As suggested in [31], several points should be considered for making makerspace and tools more accessible. Various techniques have been proposed in [32] to reduce accessibility barriers and promote inclusion. Three forms of engagement—critical, connected, and collective—support sustained engagement in the makerspace through the following items:

1.  Equipment training sessions and tours;
2.  Group events;
3.  Promoting diverse interests;
4.  Providing student leadership and volunteering opportunities [33].

### 2.3. Alternative Forms of Makerspaces

Mobile makerspace is a fast-growing manifestation of maker culture due to the high demand for makerspaces [34]. The mobile design provides more students with accessibility, engagement, and security especially for underrepresented and underserviced groups [35]. However, the overall durability and the lack of dedicated staff remain a challenge for mobile makerspaces [36].

New initiatives, such as more easily accessible e-documents and digital hardware (such as 3D printers, laser cutters, etc.) are transforming libraries into non-traditional makerspace environments [37]. As a result, librarians are trained to build partnerships that promote libraries as centres of learning and technology [31]. This movement provides more-accessible, equipped, and community-based makerspaces [32]. Successful experiences on the effective development of engaging library-based makerspaces have been reported in many university campuses. The literature discusses key considerations for equipping such spaces and training staff [38–40].

### 2.4. Makerspace Benefits

#### 2.4.1. Entrepreneurship

First-year project-based engineering courses requiring activities within makerspaces incorporate fundamental multidisciplinary engineering skills and offer excellent opportunities for Entrepreneurial-Minded Learning (EML) [41]. The maker revolution provides new opportunities for entrepreneurs, inventors, and customers to reduce product-development time [8]. Makerspaces empower student entrepreneurs to fabricate their own designs and deepen their learning experiences [17]. Open-source and Internet-of-Things (IoT) hardware/software technologies provide more affordable opportunities for product development in makerspaces [25,42].

#### 2.4.2. Self-Efficacy

Makerspaces provide an environment that allows for a blending of education, experimentation, and communication. Students using makerspaces are educated through a combination of learning, teaching, mentoring, and advising; conduct experimentations through designing, building, and fixing; and are engaging in solving communication conundrums through collaboration and participation. All these assist engineering education in nurturing confidence, increasing retention, and improving recruitment of students. Students' self-efficacy and confidence in their engineering abilities can be improved at makerspace by providing opportunities for observation, social interaction, and repetition [43]. According to [44], becoming highly involved in a makerspace improves confidence and expectations of success for conducting engineering design [1].

### 2.4.3. Creativity

As a response to declining numbers in students' preferences for STEM, young students' interest in STEM needs to be stimulated throughout schooling. The maker movement, as a key part of a third industrial revolution, provides opportunities for the creation of STEM-related artifacts through experiential learning and social interactions [45]. The makerspace ethos holds that knowledge is best acquired by a combination of theoretical (theora) and practical (praxis) approaches through building things that are concrete and sharable [12]. As a result, students' creative competency can be encouraged with DIY projects in a makerspace [46]. A hallmark of the maker movement is its DIY mindset that brings together individuals from different backgrounds where problem-solving and collaborative learning encourage creativity [47,48]. Makerspaces act as catalysts for self-efficacy, self-confidence, innovation, and creativity [37].

## 3. Engineering Programs and Makerspace at Langara College

### 3.1. Engineering Program

Langara College, as one of BC's leading undergraduate institutions, offers the "Applied Science for Engineering" two-year diploma program (http://bit.ly/2l1OYis) as well as the "Engineering Transfer" two-semester certificate program (http://bit.ly/2lj7rap). Both programs allow students to complete the equivalent of the first-year of an engineering Bachelor's program, with students in the Diploma program progressing at a slower pace. In the two-year diploma program students gain a thorough background in mathematics and natural sciences, including laboratory practice, and technical communications. They are also introduced to Canadian engineering practice, and the engineering principles of case study and design. Three project-based courses are offered as part of the two-year diploma program: "APSC 1010—Engineering and Technology in Society", "CPSC 1090—Engineering Graphics", and "CPSC 1490—Applications of Microcontrollers", with CPSC 1090 and CPSC 1490 also part of the Engineering Transfer curriculum.

### 3.1.1. APSC 1010—Engineering and Technology in Society

"APSC 1010—Engineering and Technology in Society," offered to first-term engineering students as a part of the two-year diploma program, aims to introduce students to engineering design and case studies, in addition to providing a brief survey of the history, ethics, and various disciplines of engineering (http://bit.ly/2ms2cFr). The details of this course are presented in [49].

### 3.1.2. CPSC 1090—Engineering Graphics

"CPSC 1090—Engineering Graphics" has three components: hand drawing, computer-aided design (CAD) modeling, and a group project (http://bit.ly/2mzpitE). It introduces students to the design process of engineered products, spatial visualization, and documentation through the use of proper instrumentation, free hand sketching, and CAD programming (e.g., SolidWorks). Topics covered include orthographic projections, descriptive geometry, cross-sectional views, auxiliary views, dimensioning and tolerancing, and design problem solving. The course also covers standards and conventions of engineering drawing.

### 3.1.3. CPSC 1490—Applications of Microcontrollers

"CPSC 1490—Applications of Microcontrollers" is an introductory project-based course on Arduino microcontrollers that explores the design of embedded microcontroller systems as solutions to a set of practical real-life problems. Course activities start with specific case studies and labs that apply scientific principles and technical knowledge. Activities then evolve to student-led and student-driven, team-based collaborative projects with specific practical goals. Projects require teams to document and

present their project design solutions to the entire class and to be able to illustrate key aspects of their solution using projected slides, engineering graphics, and live demonstrations (http://bit.ly/2mXSdb3).

The assessment criteria for these project-based courses are listed in Table 2.

**Table 2.** Assessment criteria for the APSC 1010, CPSC 1090, and CPSC 1490 courses.

| APSC 1010 | CPSC 1090 | CPSC 1490 |
|---|---|---|
| Final Exam<br>Midterm<br>Quizzes<br>Individual Presentation<br>Participation<br>Project Proposal<br>Progress Reports<br>Final Demonstration<br>Peer Evaluation<br>Final Report | Final Exam<br>Two Midterms<br>Quizzes<br>Assignments<br>Weekly Labs<br>Project Design<br>Simulation<br>3D Printing | Quizzes<br>3 Case Study Labs<br>Assignments<br>Participation<br>Project Proposal<br>Progress Reports<br>Final Demonstration<br>Peer Evaluation<br>Final Report |

### 3.2. Langara College Makerspace

Langara's Makerspace is both a physical resource and a conceptual framework for engagements. Rather than simply providing an accessible set of new tools and technologies, Langara's Makerspace seeks to examine how making is situated within an academic context, how intellectual rigor manifests itself therein, and how this relatively new type of space fosters creative problem-solving to pressing design and engineering challenges.

Critical making forms a keystone of our makerspace. The term critical making describes an approach to theoretically and pragmatically connect two distinct modes of engagement: critical thinking—which is typically understood as conceptually and linguistically based—and physical making, goal-based material work [50]. Rather than empowerment and engagement through proximity and skills acquisition, Langara's Makerspace seeks to foster a culture that examines what new and emergent technologies, materials, and cultures mean within a much broader critical context.

Objects and their accompanying effects and affects are inseparable. What an object is made of, who makes it, when, where, in what amount, who ultimately owns it, how it creates change in us, how well it serves its purpose, and what its life cycle is (along with a myriad of other critical questions) permeate the things around us. Making, through its participatory, collaborative, and cross-disciplinary ethos brings the ethics of objects home to the individual. Manufacturing is no longer an invisible, disembodied, and abstract process. The rise of maker culture and the rapid proliferation of inexpensive and effective technologies bring important questions to the fore as the third industrial revolution shifts conception and production toward the individual, the personal, and the local.

These egalitarian technologies have brought both new opportunities and new responsibilities. Traditional makerspaces have built considerable momentum, spreading rapidly, to a diverse array of communities and institutions. But standard methods of technological design often produce systems that lack cultural richness, emotion, and human-oriented values, and instead often overemphasize principles like efficiency and productivity which contribute to a consumer-oriented culture that overworks, overproduces, and overconsumes [51].

Langara's Makerspace (Figure 1a) explores ways to bridge the gap between how something is, or can be, made, and the broader societal questions of why it is made and what its implications may be. Some of the projects done at the makerspace are shown in Figure 1b–e.

Figure 1b shows a Langara-funded research project that examines marine plastic waste as an underutilized feedstock for the production of new objects. Open-source machine plans are being reconfigured and reconstructed to enable the shredding and injection molding of plastic waste. This technology will then be used to explore how the ability to locally remanufacture marine plastic waste can impact remote coastal communities in BC.

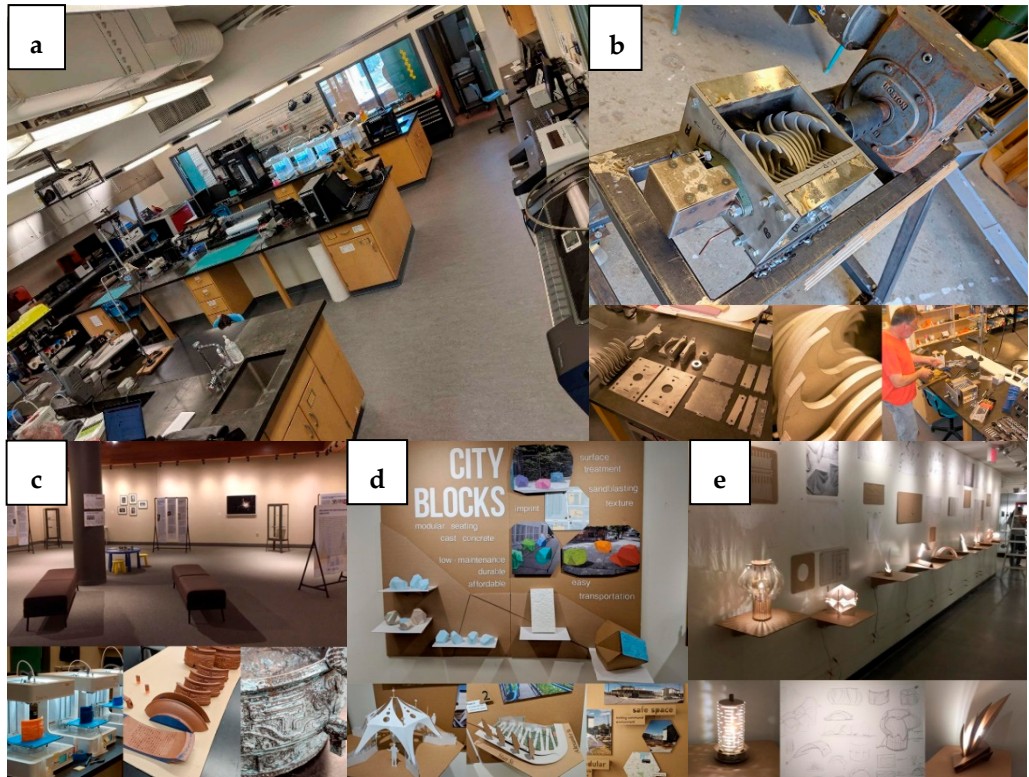

**Figure 1.** (**a**) Langara College makerspace, (**b**) Ocean plastic project (P. R), (**c**) The Prudhomme library project (H. Jessup), (**d**) Heat islands project (P. R), and (**e**) The lighting project (P. R).

Figure 1c displays a complete museum exhibition of the Prudhomme family archive of Canadian historical artifacts. Students and faculty from across disciplines created the entire exhibition (graphics, text, translations, events, and a number of "artifacts") for this national exhibition. The exhibition examined the fluid nature of "truth" and the place of credulity within an artificially-constructed narrative.

Figure 1d shows a City-of-Vancouver and CityStudio based project that addressed the twin issues of urban heat islands and the lack of sheltered gathering spaces in Vancouver's downtown eastside. Students conceived, designed, fabricated, and presented proposals. The winning entry is to be built by students and the community for $40,000.

Figure 1e depicts a 100% recyclable, laser-cut lighting kit that can be made for under $20, and be sold assembled, given away as a kit, or shared online as a set of instructions.

List of Langara's Makerspace equipment is presented in Table 3.

**Table 3.** List of Langara's Makerspace equipment.

| | |
|---|---|
| Pocket NC 5 Axis Desktop CNC Mill | Epilog Mini Laser |
| Epson Stylus Pro 4900 2D Printer | Trotec Speedy 300 Laser |
| Epson Sylus Pro 9880 2D Printer | Roland Camm-1 Servo Vinyl cutter |
| Tinkerine Dittopro 3D Printer | Romaxx CNC Cutter |
| Formlabs Form2 Resin 3D Printer | Fujifilm 4 × 6 2D Printer |
| NextEngine Ultra HD 3D Scanner | Computer Workstations |
| Recycler-Used PLA to New PLA Filament | Dry Mounting/Laminating Press |
| Makerbot Replicator Z18 3D Printer | ProtoCycler+ Plastic to Filament Recycler |
| Makerbot Replicator 2 3D Printer | Consew 225 Industrial Sewing Machine |
| Makerbot Mini 3D Printer | Electronic Workstations |

Free workshops are organized throughout the year to introduce students to the operation of makerspace equipment such as laser cutters, 3D printers, the CNC router, and the vinyl cutter. More information on Langara's makerspace can be found: https://iweb.langara.bc.ca/makerspace/.

## 4. Group Projects

Cooperative project-based learning (CPBL) improves students' technical and design knowledge. It also helps develop many soft skills, including independence, accountability, interpersonal and teamwork skills, project management, problem-solving, and self-assessment [52]. Many of the projects are selected based on the mixed-ability maker culture which entails making useful things for people with disabilities [30]. In CPSC 1090, students were assigned practical real-life projects based on projects proposed by Tetra Society of North America (https://www.tetrasociety.org/)—a volunteer-based organization focusing on finding solutions to overcome environmental barriers faced by people with disabilities. In APSC 1010 and CPSC 1490, students are also encouraged to define their own projects with the appropriate guidance from the instructor based on a real-life problem. The following steps are applied by the project teams in the engineering design process [36]:

1. Define the problem;
2. Identify constraints;
3. Brainstorm;
4. Select a solution;
5. Prototype;
6. Test;
7. Iterate and Improve.

Students in APSC 1010, CPSC 1090 and CPSC 1490 apply three stages for the group projects.

### 4.1. Team Forming and Project Selection

In both APSC 1010 and CPSC 1490 team formation is done through self-selection, based on the students' knowledge of each other's educational background, specialized knowledge, and skills. Conversely, as CPSC 1090 mimics a real-world design project within an existing engineering company, students are randomly assigned to their groups. This creates opportunities to build teamwork, collaboration, and interpersonal skills. Random assignment also helps ensure comparable groups and minimizes the influence of individual characteristics, such as previous experience with the CAD program. Ideally, all three courses would assign students based on their interests and skillsets. This will be implemented shortly in all three courses [53].

### 4.2. Progress Reports and Makerspace Meetings

The teams are expected to meet on a weekly basis to discuss problem-solving methodologies. Instructor input is offered when necessary to provide research literature and differing points of view. To avoid poor performance due to dysfunctional teams, progress reports and peer evaluation techniques are utilized. In the last four weeks of the course, students have access to Langara College's Makerspace which expands experiential learning opportunities and allows students to gain a skillset outside of the traditional classroom.

### 4.3. Final Presentation and Evaluation

At the end of the term, groups deliver a 15-min PowerPoint presentation to the class followed by five minutes of questions. In this presentation, students explain the design processes and challenges in order to defend their design choices. Work division, time-management, and budget are also discussed in detail. Then, they show a demo of their final prototype to the class.

### 4.3.1. Peer Evaluation

Each team member completes a scoring matrix, rating every team member's total contributions to the project. These matrices are averaged amongst all team members and used to weight the project mark according to individual contributions. The criteria for peer evaluation are:

1. Punctuality and regular attendance in group meetings;
2. Meaningful contributions to group discussions;
3. Completing group assignments on time;
4. Preparing work in a quality manner;
5. Demonstrating a cooperative and supportive attitude;
6. Significant contributions to the success of the assignment.

### 4.3.2. Student Projects

In APSC 1010 and CPSC 1490 project teams select an open-ended topic of their own interest with achievable goals. The instructor provides the necessary research resources and guides the groups in tailoring their topics so that the work is authentic, based on real-life problems, low-budget, and challenging. Teams propose their topic in a presentation to receive feedback from their peers and the instructor [49].

Students used the makerspace as a location for their group meeting and they utilized different available equipment such as 2D and 3D printers, laser and vinyl cutters, Arduino and Raspberry-Pi microcontrollers, electric circuitry, sensors and actuators, and metal/woodwork hand tools. Moreover, makerspace provided an effective training environment through mentor supervision, formal workshop sessions, and informal peer education. Figure 2 shows some of the project results delivered for APSC 1010 and CPSC 1490.

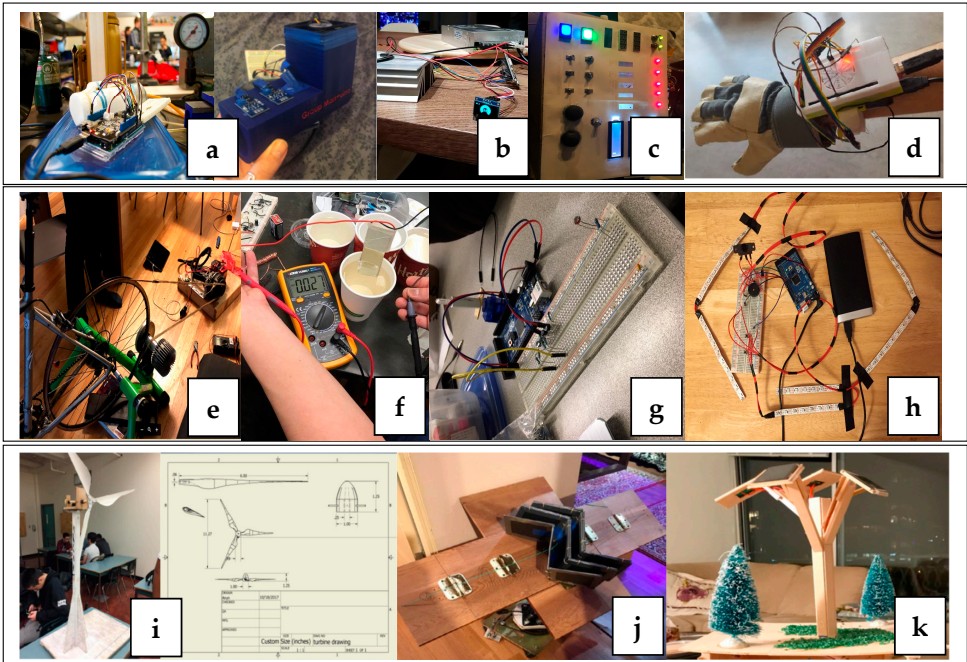

**Figure 2.** APSC 1010 and CPSC 1490 sample projects: (**a**) Gas-monitoring system, [54], (**b**) Arduino-based coaster, (**c**) Kerbal space game controller, (**d**) Smart glove prototype for the physically impaired [55], (**e**) Pedal powered electric generator, (**f**) Electricity generation using Peltier tile, (**g**) Prototype for automatic blind opener, (**h**) Turn-signal bike jacket prototype, (**i**) Wind turbine prototype, (**j**) Origami-inspired portable solar panel [56], and (**k**) Solar-tree prototype.

In CPSC 1090, a representative from the project partner (Tetra Society) is also invited to attend the presentations and test the final product (see Figure 3 for the cup holder project testing). The feedback from the stakeholder is well received by the students, and leads to discussions about potential improvement and optimization of the proposed solution. Work division, time-management, and budget are also discussed among the students in detail.

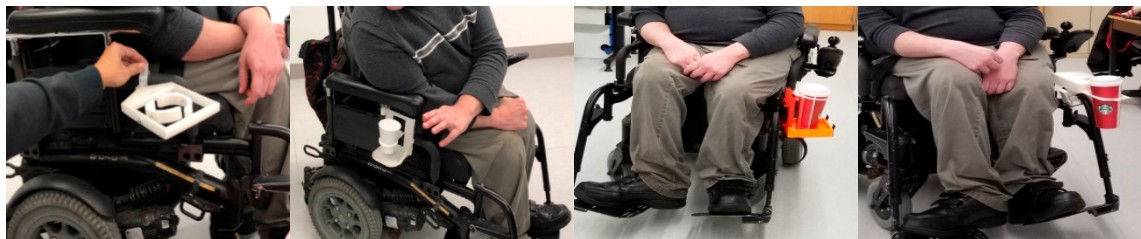

**Figure 3.** Four different 3D designs of a folding cup holder for wheelchair.

## 5. Student Survey

Table 4 presents a list of projects completed by APSC 1010 and CPSC 1490 students. Although recent studies question the adequacy of Student Evaluation of Teaching (SET) as a solo measure of teaching effectiveness, SET scores are still the most common evaluation method. Research shows that to improve the efficiency of SET, students' comments should be considered carefully in the review process in parallel with classroom observations [57]. Utilization of shorter surveys and communication of changes implemented based on student feedback are two motivators for higher student participation in SET [58]. Averages in smaller samples/scales are more susceptible to misleading results than averages in larger samples/scales [59]. To evaluate the effectiveness of using makerspace and group projects, anonymous voluntary surveys were conducted at the end of both courses. These surveys consisted of a series of close-ended questions asking the students, on a scale of 0 to 10, to what extent they agreed with a set of fixed statements. The surveys provide additional measures for evaluating the effectiveness of makerspace in addition to final projects quality and students' participation monitored by the instructors. Tables 5 and 6 show the questions and the percentage of students providing a number greater than or equal to 7. More comprehensive results including number of participants, average, and standard deviation are provided in Figures 4 and 5. The results confirm the effectiveness of these courses in achieving its goals. This process also verifies how project-based courses and the use of makerspaces solidified engineering concepts for students.

**Table 4.** List of group projects for APSC 1010 and CPSC 1490.

| | |
|---|---|
| Wind-Powered Buildings | Smart Robot Car |
| Peltier effect | Pedal-powered generator |
| Automated blind system | Water clock model |
| Solar tree | Portable solar panel systems |
| Self-sufficient service station | Smart wrist assist |
| Water consumption tracking | Eye-tracking interaction system |
| Turn-signal jacket | Smart glove |
| Automatic irrigation system | Automatic page flipper |
| Algae-based air filter | Smart coaster |
| Dual-purpose solar panel | IoT gas monitoring interface |
| Emergency power charger | Robotic arm |
| Solar-powered farm | Rotating aeroponics growth unit |
| Sustainable bridge | Kerbal space game controller |

**Table 5.** Survey questions and percentage of responses with answers ≥ 7/10 for APSC 1010.

| Evaluation Question | Responses ≥ 7/10 |
|---|---|
| Q1. Did the presentations scheduled for the course improve your presentation skills? | 88.5% |
| Q2. Did the group project improve your project-management skills? | 84.6% |
| Q3. Did the group project improve your communication/interpersonal skills? | 84.6% |
| Q4. Did the group project help you improve your time-management skills? | 73.1% |
| Q5. How satisfied were you with accommodations provided for the group project? | 78.4% |
| Q6. Were the scheduled progress reports helpful in making sure you are on track? | 73.1% |
| Q7. Did this course help you realize which engineering field you are interested in? | 69.2% |

**Table 6.** Survey questions and percentage of responses with answers ≥ 7/10 for CPSC 1490.

| Evaluation Question | Responses ≥ 7/10 |
|---|---|
| Q1. This course has increased my interest in engineering or computer science. | 86.4% |
| Q2. The hands-on experiences increased my interest in eng. or computer science. | 70.4% |
| Q3. I learned practical technical concepts as a result of working on the group project. | 79.5% |
| Q4. The labs and makerspace provide an appropriate environment for this course. | 84.1% |

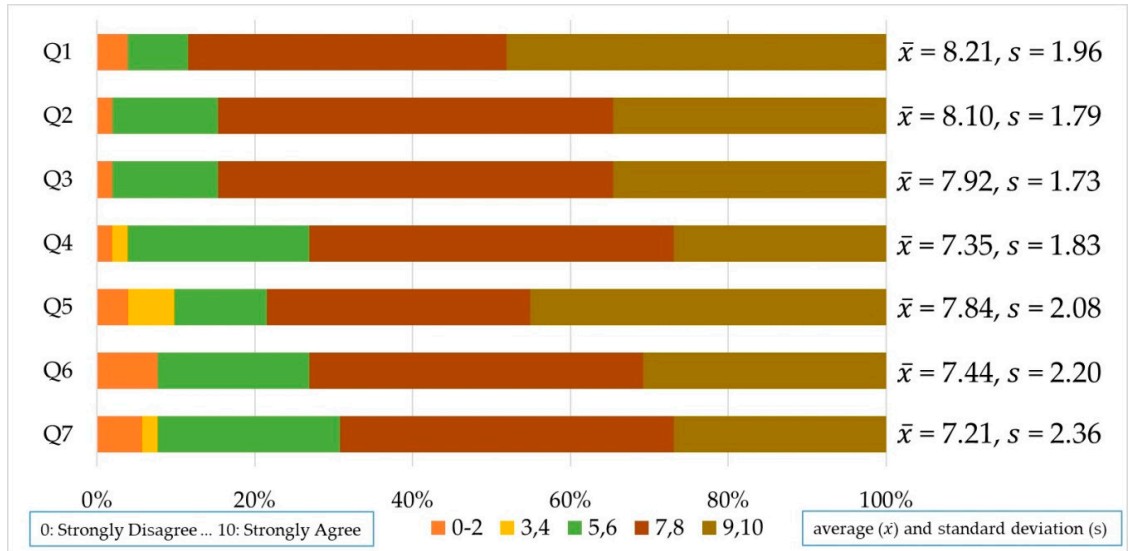

**Figure 4.** APSC 1010 survey results (N = 61).

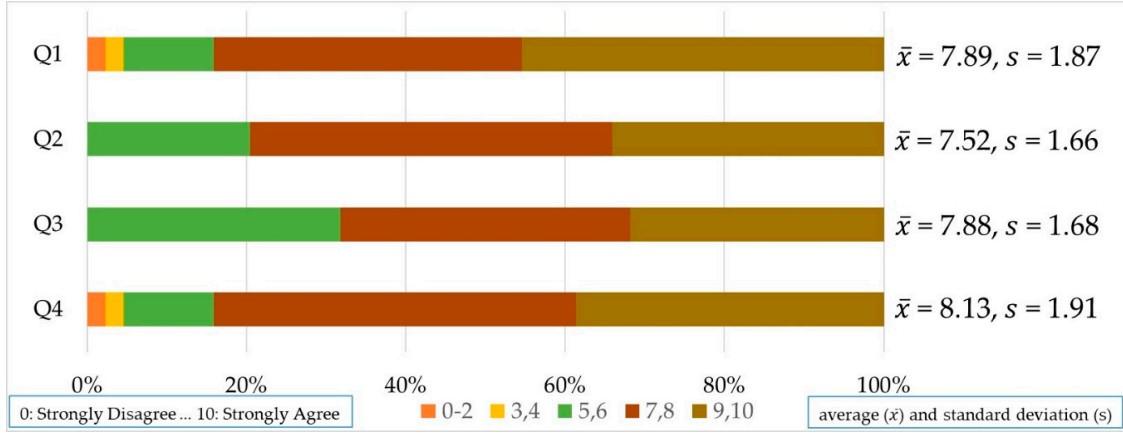

**Figure 5.** CPSC 1490 survey results (N = 57).

## 6. Conclusions

This paper provides a review of the history and benefits of makerspace and the maker culture in postgraduate studies with a focus on engineering disciplines. Different considerations involved in creating and utilizing a makerspace are discussed and various forms of makerspaces are briefly introduced.

Three project-based courses offered as part of Langara's two-year engineering diploma program ("APSC 1010—Engineering and Technology in Society", "CPSC 1090—Engineering Graphics", and "CPSC 1490—Applications of Microcontrollers) and their group project component are introduced in detail. The paper then demonstrates how the use of makerspaces can potentially improve students' experiences in these first-year engineering-design courses. Instructors observed different impacts of a makerspace on first-year engineering students such as:

1. Improving students' communication, collaboration, and teamwork skills;
2. Developing problem-solving and investigation skills;
3. Nurturing entrepreneurial, leadership, and management skills;
4. Increasing students' self-confidence;
5. Providing opportunities for hands-on and technical experience;
6. Demonstrating manufacturing/prototyping methods and the challenges/limitations involved.

Many samples of successful student projects implemented at makerspace were presented. Based on instructors' observation, project qualities, and student evaluations, the makerspace and project-based courses appear to highly improve students' learning experiences in engineering courses.

Due to the fast-changing industry and increasing need to create multidisciplinary teams in the current digital economy, there is a high demand for formal and effective education of specialists and leaders [60]. Makerspaces can provide a perfect environment and platform to facilitate both theoretical and practical trainings. Other prospects for using makerspace to train instructors, technicians, managers, and entrepreneurs are under development through continuing-study programs at Langara College.

In this study, based on the quality of final projects, instructors' observations, and student evaluations, early introduction of makerspaces was identified as a promising factor to improve students' active learning over an engineering program. Since Langara's makerspace has been in operation only since 2017, the statistical data verifying its role in improving engineering students' retention is not available at the moment. As a future novel work, the statistical data gathered from different institutions over several years can be utilized to analyze the impacts of makerspaces in students' retention, GPAs, and successful graduation.

**Author Contributions:** P.T. was the instructor for APSC 1010 and CPSC 1490, and he wrote the original draft and revised version of this paper; P.R. is the coordinator of Langara College makerspace and he wrote the Section 3.2 of this paper; S.M. was the instructor for CPSC 1090 and she wrote the corresponding Sections 3.1 and 4.3 of this paper. All authors have read and agreed to the published version of the manuscript.

**Funding:** This research received no external funding.

**Acknowledgments:** Pooya Taheri would like to thank Steve Whitmore, Haida Antolick, Amy Yeung, Katie Mak, and Csilla Tamás for their valuable feedback. Sirine Maalej would like to thank Eric Molendyk for being a partner in CPSC 1090 projects.

**Conflicts of Interest:** The authors declare no conflict of interest.

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
