# Peer review of "Makerspaces in First-Year Engineering Education"

_education, doi:10.3390/educsci10010008_

Round 1

Reviewer 1 Report

In my opinion the paper is more an analysis of a study case (Langara makerspaces) than an experimental project.

The document lacks an introduction where to put very important aspects such as:

Contextualization of the makerspace project The limits of previous research that justify the need of the article. Motivation of the choice of research. Method or design Description of the main results and main findings. Please consider adding a purpose statement and research questions. 

The abstract is more a short introduction than an abstract, because it contains some ideas that should not be in an abstract. It should be a condensed paper version.

The acronyms should be introduced in their first usage. The author use BC, APSC and CPSC  with no introduction.

The authors claim section 1.3 to be the literature review. However, sections 1 and 2 actually contains the literature review. The structure should be clarified.

The authors explain makerspaces in many detail, but the specific details of the Langara use case are not detailed, while these are the most interesting part of the context. The images clarify much about the process of creation.

The conclusions are short and very poor. I’m sure that the project allows for much more than 6 lines.

It would also be interesting to suggest future research about makerspaces.

Author Response

We would to thank the reviewer for providing very constructive feedback. We have implemented the reviewers' comments to the best of our ability to improve the quality of the paper. Please see our response to your comments below:

1. The document lacks an introduction where to put very important aspects such as Contextualization of the makerspace project The limits of previous research that justify the need of the article. Motivation of the choice of research. Method or design Description of the main results and main findings. Please consider adding a purpose statement and research questions. 

Response: We agree with the reviewer. In the revised version of the paper section 1.4 of the introduction has been improved to clearly specify the goal and motivation of this paper.

2. The abstract is more a short introduction than an abstract, because it contains some ideas that should not be in an abstract. It should be a condensed paper version.

Response: We agree with the reviewer. The abstract has been improved as a condensed version of the paper.

3. The acronyms should be introduced in their first usage. The author use BC, APSC and CPSC  with no introduction.

Response: We corrected this issue in the revised version.

4. The authors claim section 1.3 to be the literature review. However, sections 1 and 2 actually contains the literature review. The structure should be clarified.

Response: We agree with the reviewer. To avoid confusing the readers, we renamed section 1.3.

5. The authors explain makerspaces in many detail, but the specific details of the Langara use case are not detailed, while these are the most interesting part of the context. The images clarify much about the process of creation.

Response: We agree with the reviewer. In sub-section 4.3.2, we added a paragraph detaining how makerspace was used in student projects.

6. The conclusions are short and very poor. I’m sure that the project allows for much more than 6 lines.

It would also be interesting to suggest future research about makerspaces.

Response: We 100% agree with the reviewer. The conclusion has been rewritten to cover the outcome of this paper. We have added future project suggestions at the end of the conclusion section.

We would to thank the reviewer again for very helpful comments.

Reviewer 2 Report

This is a really good paper that provides a comprehensive overview of the  Makerspace capabilities. The following comments may allow authors to present the results of research in a more convincing manner.

Sometimes the presentation style is more like a textbook than an article. I suggest the authors analyze where it is possible to direct the presentation style towards discussion, and, on the contrary, present some pieces in a more systematic way. For example, the text in sections 3.1.1-3.1.3 can be presented in the form of a comparative table.

Section 5 deals with survey results having a high degree of representativeness. However, I would like to see behind these results something more than just the conclusion that the use of Makerspace is effective in education. In this regard, the field of further research by the authors is of interest. This can be said in the Conclusion, which so far is very formal and seems to be the weakest point of the paper. Perhaps there are prospects for using the Makerspace to train other specialists, for example, managers or entrepreneurs, since in the digital economy there is an increasing need to create multidisciplinary teams already at the training stage. (For more details see, for example, Gitelman, L.; Kozhevnikov, M.; Ryzhuk, O. Advance Management Education for Power-Engineering and Industry of the Future. Sustainability 201911, 5930).

A small question for understanding: why is the use of makerspace considered only in the first year of study? Do I understand correctly that this tool can actually be used throughout the entire training period? I would like to clarify why the authors focus only on the first year of engineering education?

Finally, two small notes on the paper design: 1) I could not find a reference to Table 1 in the text, 2) the purpose of the abbreviations in Fig. 3 is unclear - did the authors decide to put the initials of developers or project supervisors? Are these abbreviations really necessary?

Author Response

We would to thank the reviewer for providing very constructive feedback. We have implemented the reviewers' comments to the best of our ability to improve the quality of the paper. Please see our response to your comments below:

1. Sometimes the presentation style is more like a textbook than an article. I suggest the authors analyze where it is possible to direct the presentation style towards discussion, and, on the contrary, present some pieces in a more systematic way. For example, the text in sections 3.1.1-3.1.3 can be presented in the form of a comparative table. 

Response: We agree with the reviewer. We modified sections 3.1.1-3.1.3 as well as other parts of the paper as per reviewer's suggestion.

2. Section 5 deals with survey results having a high degree of representativeness. However, I would like to see behind these results something more than just the conclusion that the use of Makerspace is effective in education. In this regard, the field of further research by the authors is of interest. This can be said in the Conclusion, which so far is very formal and seems to be the weakest point of the paper. .

Response:We 100% agree with the reviewer. The conclusion section has been rewritten to cover the outcome of this paper. We have added future project suggestions at the end of this section.

3. Perhaps there are prospects for using the Makerspace to train other specialists, for example, managers or entrepreneurs, since in the digital economy there is an increasing need to create multidisciplinary teams already at the training stage. For more details see, for example, Gitelman, L.; Kozhevnikov, M.; Ryzhuk, O. Advance Management Education for Power-Engineering and Industry of the Future. Sustainability 2019, 11, 593.

Response: We thanks the reviewer for a very constructive suggestion and bringing this publication to our attention. We have added the suggestion to the conclusion section and added a citation to the mentioned journal paper.

4. A small question for understanding: why is the use of makerspace considered only in the first year of study? Do I understand correctly that this tool can actually be used throughout the entire training period? I would like to clarify why the authors focus only on the first year of engineering education?

Response: 

We would like to thank the reviewer for addressing this confusing point in our paper.

The use of makerspace can be useful in every stage of an engineering program. However, Langara College only offers the “Applied Science for Engineering” two-year diploma program as well as the “Engineering Transfer” two-semester certificate program. Both programs allow students to complete the equivalent of the first-year of an engineering Bachelors program, with students in the Diploma program progressing at a slower pace. We do not have a formal 4-year Bachelor program in our college at this time. 

In this paper, we propose that early introduction of makerspace can be a promising factor to improve students’ active learning throughout an engineering program.

We tried to emphasize on these points in the revised version of the paper.

5. I could not find a reference to Table 1 in the text.

Response: Thank you for pointing this out. We corrected this issue.

6. the purpose of the abbreviations in Fig. 3 is unclear - did the authors decide to put the initials of developers or project supervisors? Are these abbreviations really necessary?

Response: Thank you for your question. As added in the revised version of the paper, we added the initials of the project developers below their corresponding project's picture. We wanted to give credit where it is due, however we were not allowed by the college to add students' full names.

We would to thank the reviewer again for very helpful comments.

Reviewer 3 Report

the paper is well written and the basic sections (introduction, conclusion, literature cited, etc.) are adequate. The author wrote subheadings and they were relevant to the text as they clarified the sections of the text. The material order is easy to follow. The author's writing style was clear. However, there were some spelling mistakes. 
The review section is adequate. In the conclusion section the authors should discuss the paper more.

Author Response

We would to thank the reviewer for providing very constructive feedback. We have implemented the reviewers' comments to the best of our ability to improve the quality of the paper. Please see our response to your comments below:

1. There were some spelling mistakes. . 

Response: Thank you for bringing this issue to our attention. We proofread the paper multiple times to identify and correct the spelling and grammar mistakes in the revised version.

2. In the conclusion section the authors should discuss the paper more.

Response:We 100% agree with the reviewer. The conclusion section has been rewritten to cover the outcome of this paper.

We would to thank the reviewer again for very helpful comments.

Reviewer 4 Report

The article describes an experience carried out by the authors. They report a survey filled by the participant students, but the numbers are very low (about 60 students in each survey). To get reliable results the sample should be improved: there should be more students engaged in the workshops described. The experience is good ande the described workshops very nice but you should test them with a bigger numer of students.

Author Response

Thanks for your comments.

Reviewer 5 Report

Authors should specify why a 1-10 scale was chosen. Diferent scales might provide different conclusions.

Also conclusions should be improved with the limitations of the study. 

The final conclusion can be only suported if there is statisticas significance, which mean/standard and stdev. cannot provide.

Either conclusions reflect this or the lack of significance is limited so far.

Author Response

We would to thank the reviewer for providing very constructive feedback. We have implemented the reviewers' comments to the best of our ability to improve the quality of the paper. Please see our response to your comments below:

1. Authors should specify why a 1-10 scale was chosen. Diferent scales might provide different conclusions.

Response: We agree with the reviewer. We added three references discussing the student evaluation of teaching and discussed their finding briefly in section 5. According to these literature, Although recent studies question the adequacy of Student Evaluation of Teaching (SET) as a solo measure of teaching effectiveness, SET scores are still the most common evaluation method. Averages in smaller samples / scales are more susceptible to misleading results than averages in larger samples / scales. 

2. Also conclusions should be improved with the limitations of the study. 

Response: We 100% agree with the reviewer. The conclusion has been rewritten to cover the outcome of this paper. We have added future project suggestions at the end of the conclusion section.

3. The final conclusion can be only suported if there is statisticas significance, which mean/standard and stdev. cannot provide. Either conclusions reflect this or the lack of significance is limited so far.

Response: We 100% agree with the reviewer again. As added in the conclusion section, in this study, based on the quality of final projects, instructors’ observations, and student evaluations, early introduction of makerspace was identified as a promising factor to improve students’ active learning throughout an engineering program. Since the Langara’s Makerspace has been in operation only since 2017, the statistical data verifying its role in improving engineering students’ retention is not available at the moment. As a future novel work, the statistical data gathered from different institutions throughout several years can be utilized to analyze the impacts of makerspace in students’ retention, GPAs, and successful graduation.

We would to thank the reviewer again for very helpful comments.

Round 2

Reviewer 1 Report

Once the reviews have been analyzed, the paper is suitable for publication.

Reviewer 5 Report

Paper is now acceptable for publication. Good work.